# Effect of Long-Term Cropping Systems on the Diversity of the Soil Bacterial Communities

Zoltán Mayer [1,†], Zita Sasvári [1,†], Viktor Szentpéteri [1], Beatrix Pethőné Rétháti [1], Balázs Vajna [2] 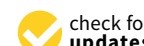 and Katalin Posta [1,3,*]

1    Institute of Genetics, Microbiology and Biotechnology, Szent István University, HU-2100 Gödöllő, Hungary; mayer.zoltan@mkk.szie.hu (Z.M.); sazyta@gmail.com (Z.S.); szviki213@gmail.com (V.S.); pethone.rethati.beatrix@mkk.szie.hu (B.P.R.)

2    Department of Microbiology, Eötvös Loránd University, HU-1117 Budapest, Hungary; vajna.balazs@ttk.elte.hu

3    Institute of Biotechnology and Food Technology, Industrial University of Ho Chi Minh City, Ho Chi Minh City VM-71406, Vietnam

*    Correspondence: posta.katalin@mkk.szie.hu; Tel.: +36-28-522-000 (ext. 2105)

†    The authors contributed equally to this article.

**Abstract:** Soil microbial communities are involved in the maintenance of productivity and health of agricultural systems; therefore an adequate understanding of soil biodiversity plays a key role in ensuring sustainable use of soil. In the present study, we evaluated the influence of different cropping systems on the biodiversity of the soil bacterial communities, based on a 54-year field experiment established in Martonvásár, Hungary. Terminal restriction fragment length polymorphism (T-RFLP) fingerprinting technique was used to assess soil bacterial diversity and community structure in maize monoculture and three different crop rotations (maize–alfalfa, maize–wheat and the maize–barley–peas–wheat Norfolk type). No differences in richness and diversity were detected between maize monoculture and crop rotations except for the most intense rotation system (Norfolk-type). Although the principal component analysis did not reveal a clear separation between maize monoculture and the other rotation systems, the pairwise tests of analysis of similarity (ANOSIM) revealed that there are significant differences in the composition of bacterial communities between the maize monoculture and maize–alfalfa rotation as well as between wheat–maize and Norfolk-type rotation.

**Keywords:** long-term; crop rotation; monoculture; terminal restriction fragment length polymorphism (T-RFLP) fingerprint; soil bacterial community

## 1. Introduction

Soil microorganisms play an important role in many soil processes, including carbon and nitrogen cycling, nutrient acquisition by plants, production of soil aggregates and the conversion of plant residues to soil organic matter [1]. Furthermore, the presence of antagonistic microbes and the diversity of soil microbial communities contribute substantially to the resistance and resilience of ecosystems to biotic disturbance and stresses [2–4]. The composition of soil communities is affected by several factors like soil properties, seasonality or management practices [5]. Agricultural land management practices are one of the most significant anthropogenic activities that greatly alter soil characteristics, including physical, chemical and biological properties [6]. Moreover, these activities are often found to decrease soil microbial diversity [7]. Long-term fertilization and cultivating methods have a significant effect on soil microbial community composition. The relative abundance of several bacterial genera

alters with the use of fertilizer, also different fertilizer types induce different changes [8,9]. Not only bacterial, but also fungal communities are altered by fertilizer use, method of tillage and crop rotations. Inorganic fertilizers are proven to decrease soil fungal diversity [10], while the use of organic fertilizers and crop rotation affect the abundance of different fungal genera [11].

Due to the complexity of the dynamics that regulate soil bacterial communities, the mechanisms underlying the effects of crop rotation are still poorly understood [12]. In general, different crop rotation systems, both the length of the rotation and the plant species included in the rotation, have big influences on soil characteristics and crop benefits [13,14]. Although there is extensive research on soil microbial diversity, relatively few studies have been undertaken to evaluate the lasting impact of crop rotation on bacterial diversity and community structure by means of terminal restriction fragment length polymorphism (T-RFLP) [15,16]. Despite the limitations of this technique, it provides a high-throughput, cultivation-independent, reproducible approach to rapidly describe and compare bacterial communities from a large number of samples [17,18]. In addition, data collection from a 54-year field experiment enables us to evaluate the long-term effect of crop rotation practices on soil bacterial communities. Furthermore, sampling carried out on parcels free from any kind of fertilization allows us to identify the effect of crop rotations without any interference from a significantly biasing agronomical factor.

Therefore, the aims of this study were (i) to determine whether long-term maize monocropping causes differences in soil parameters and, moreover, in richness, diversity and structure of bacterial communities, compared to rotation systems; (ii) to further assess whether there are differences among the different cropping systems; and (iii) to investigate which environmental factors are most strongly correlated with changes in bacterial communities.

## 2. Materials and Methods

### 2.1. Study Site and Soil Sampling

The sampling site was located at Martonvásár (47°21′ N, 18°49′ E), Hungary, where the experimental trials were established in 1958 on a humus-rich loam of the chernozem type soil with forest residues. The climate of the region is classified as continental: the mean annual temperature and precipitation between 1958 and 2018 were 10.6 °C and 539 mm, respectively. The humus content was measured after digestion with potassium dichromate and cc. sulfuric acid by a spectrophotometer (U-2900, Hitachi, Japan). Soil pH was measured with a digital pH meter (HQ411D, Hach-Lange, Loveland, CO, USA) in KCl. Calcium carbonate ($CaCO_3$) content was determined through volumetrically released $CO_2$ of dry soil samples treated with 10% HCl. The available nitrate (N) content was measured by the method of Felföldy [19], phosphorus ($P_2O_5$) and potassium ($K_2O$) content measurements were carried out according to Egnér et al. [20] using ammonium-lactate for soil extraction.

The crop rotation experiment was a two-factorial split-plot with four replications. The main plots (49 m × 5 m = 245 $m^2$) consisted of seven crop sequences and the subplots (7 m × 7 m = 49 $m^2$) consisted of five fertilizer treatments, in a randomized design that received the same treatment year after year [21]. Crops were harvested in the end of October, followed one month later by a conventional tillage of 20 cm depth. Weeds and insects were controlled by pesticide treatments in all plots as described in detail by Magurno et al. [22]. Four cropping systems were chosen for sampling: maize monoculture (CR1) as a control, 3 years alfalfa and 5 years maize (CR3), 2 years wheat and 2 years maize (CR5) and Norfolk-type rotation of maize, spring barley, peas and wheat (CR7). At the sampling time, wheat was grown in case of the Norfolk type, and maize was grown in all the other rotations. Two subplots (two replications P1 and P2) were assigned to each rotation system and five samples (soil cores of 5 cm diameter and 25 cm length were collected, the top 5 cm of the cores was removed and the rest were mixed thoroughly) were randomly collected from each subplot on the 30th of June 2012, choosing only those without fertilization (control subplots in the fertilizer treatments). In total, 40 soil

samples were collected (4 cropping systems × 2 subplots × 5 soil cores). The soil samples were stored in separate plastic bags at −20 °C until processing.

### 2.2. DNA Extraction and 16 rDNA T-RFLP Analysis

DNA extraction from 0.5 g of soil samples was performed using a FastDNA SPIN Kit for Soil (Q-BIOgene, Heidelberg, Germany) according to the manufacturer's protocol. Phusion Bacterial Profiling Kit from Thermo Fisher Biosciences (Waltham, MA, USA) was used to amplify polymorphic segments of the bacterial 16S ribosomal RNA genes by PCR using 8F (5′ AGA GTT TGA TCC TGG CTC AG 3′—blue) and 926R (5′ CCG TCA ATT CCT TTR AGT TT 3′—yellow) fluorescently labelled (6-FAM and NED) universal primers. To obtain molecular fingerprints after amplification, 16S rDNA amplicons were double digested with restriction enzymes *Msp*I (C′CGG) and *Hin*P1I (G′CGC). The cleaved products were then separated and detected on a Model 3100 ABI PRISM® Genetic Analyzer (Applied Biosystems, Foster City, CA, USA). The electrophoretic profiles of the samples were determined using Peak Scanner 1.0 software to calculate fragment sizes by comparison with a GeneScan LIZ1200 size standard (Applied Biosystems, Foster City, CA, USA).

### 2.3. Statistical Analysis

T-RFLP data from Peak Scanner 1.0 was exported and processed and analyzed with T-REX software (http://trex.biohpc.org/; [23]). The T-RFLP data were subjected to noise filtration based on standard deviation (multiplier = 1.5) of peak area and aligning terminal restriction fragments (T-RFs) in all samples with 1 bp clustering threshold. The relative abundance of a detected T-RF within a given T-RFLP profile was calculated. The normalized data matrix was used for multivariate statistical analysis of T-RFLP data using the PAST (PAleontological STatistics v3.05) software package [24]. In order to visualize relationships among the T-RFLP profiles of bacteria, principal component analysis (PCA) was performed. Based on the Bray–Curtis values, analysis of similarity (ANOSIM) was used to test statistically whether there is a significant difference between bacterial communities of different cropping systems. Calculation of biodiversity indices was also performed using PAST. The soil physicochemical properties and the biodiversity indices were statistically analyzed with the R Statistical Software 3.3.1 [25]. For analysis of the soil physicochemical properties, one-way ANOVA and Dunnett's post-hoc test were used to compare the various rotation systems to monoculture. The biodiversity indices were analyzed with one-way ANOVA, and Tukey's post-hoc test was used to compare rotation systems. "Envfit" script of R program was used to fit environmental vectors onto PCA ordination and to assign the statistical significance of canonical correlation coefficients of this fitting by means of random permutations of the data.

## 3. Results and Discussion

### 3.1. Soil Properties

The major characteristics of soils originating from different treatments are summarized in Table 1. Based on the results of soil analyses, the significance of the *F* values was also calculated. According to the Dunnett's post-hoc test, there was a significant effect ($p < 0.05$) of crop rotations on soil humus content (CR3, $p = 0.000365$; CR5, $p = 0.000247$; CR7, $p = 0.000174$), $P_2O_5$ (CR3, $p = 0.0409$; CR5, $p = 0.0300$) and ($NO_3^- + NO_2^-$) –N concentration (CR3, $p = 0.0308$) compared to maize monoculture. Long-term monoculture and crop rotations caused no significant differences in soil pH. Soil pH is important in the control of abiotic factors, such as nutrient and carbon availability [26], and biotic factors, such as fungi and bacteria composition. Moreover, it is also suitable for predicting microbial diversity, as shown by Jiang et al. [27]. Our finding that maize monoculture has the lowest humus content is in accordance with the results of an earlier study of Berzsenyi et al. [21]. Humus affects soil properties: it retains moisture and allows for better drainage by loosening the soil. Long-term maize monoculture system significantly ($p < 0.001$) reduced the humus content, which correlates with decreased bacterial

diversity, as found in the work of Zhao et al. [28]. Higher concentrations of humus could increase the amount of essential absorbable nutrients for plant growth [29] and enhance the number of soil microorganisms [30].

**Table 1.** The major chemical properties of the experimental soils in different treatments.

| Cropping System | Humus (%) [1] | pH (KCl) [1] | CaCO$_3$ (%) [1] | P$_2$O$_5$ (mg kg$^{-1}$) [1] | K$_2$O (mg kg$^{-1}$) [1] | (NO$_3^-$+NO$_2^-$)–N (mg kg$^{-1}$) [1] |
|---|---|---|---|---|---|---|
| CR1 (mono) | 2.31 ± 0.05 | 5.65 ± 0.14 | 0.29 ± 0.01 | 85.85 ± 9.22 | 259.83 ± 19.96 | 4.22 ± 1.63 |
| CR3 (rotation) | 2.72 ± 0.05 *** | 5.13 ± 0.10 | 0.29 ± 0.03 | 61.27 ± 6.37 * | 244.50 ± 13.09 | 9.57 ± 2.01 * |
| CR5 (rotation) | 2.74 ± 0.05 *** | 5.68 ± 0.06 | 0.16 ± 0.02 | 59.82 ± 3.92 * | 275.33 ± 14.27 | 5.50 ± 0.71 |
| CR7 (rotation) | 2.75 ± 0.08 *** | 5.95 ± 0.14 | 0.30 ± 0.09 | 73.10 ± 5.81 | 287.17 ± 8.48 | 4.50 ± 0.52 |
| One-way ANOVA [2] | F = 11.99, $p$ = 0.0001 *** | F = 2.974, n.s. | F = 2.067, n.s. | F = 3.366, $p$ = 0.039 * | F = 1.630, n.s. | F = 3.295, $p$ = 0.0416 * |
| Dunnett's post-hoc test | CR3: $t$ = 4.74, $p$ = 0.000365 *** CR5: $t$ = 4.89, $p$ = 0.000247 *** CR7: $t$ = 5.04, $p$ = 0.000174 *** | | | CR3: $t$ = −2.63, $p$ = 0.0409 * CR5: $t$ = −2.78, $p$ = 0.0300 * | | CR3: $t$ = 2.77, $p$ = 0.0308 * |

[1] Values are presented as the mean ± SE. [2] * Significant at $p < 0.05$, *** significant at $p < 0.001$, n.s. = nonsignificant, according to one-way ANOVA and Dunnett's post-hoc test (df = 3; $p < 0.05$). CR1, maize monoculture; CR3, 3 years alfalfa and 5 years maize; CR5, 2 years wheat and 2 years maize; CR7, Norfolk-type rotation of maize, spring barley, peas and wheat.

Liu et al. [31] found that increasing the crop rotation of legumes could release nutrients more quickly, leading to higher nitrogen (N) and potassium (K) content in plants and lower available phosphorus (P) content in soil. The increased N concentration in the soil of the alfalfa and maize rotation system is due to the activities of nitrogen-fixing bacteria and phosphate solubilizing microorganisms which are associated with the cultivation of alfalfa [32]. Our results showed that rotation increased the amount of N and organic matter in soil; the increase of these factors also promotes microbial diversity [33,34].

### 3.2. Richness and Diversity of Bacterial Communities

The biodiversity of the total soil bacterial communities for each trial were analyzed by T-RFLP. Samples collected in June directly before flowering should have a relatively robust and stable bacterial community, as found by Wang et al. [35] and Ishaq et al. [36]. A total of 99 different T-RFs with sizes ranging from 51 to 554 base pairs (bp) were detected across all sites. The average number of T-RFs in the dataset (average T-RF richness) was 48.40 (F = 4.547, df = 3, $p$ = 0.0153). Sample heterogeneity, also known as beta diversity [37], was rather low (1.04). The highest average number of T-RFs (55.50 ± 1.09) was found in the alfalfa–maize rotation, followed by the wheat–maize rotation (54.83 ± 2.39). They were both significantly higher ($p < 0.05$) than in Norfolk-type rotation (33.66 ± 1.85), but not significantly different compared to maize monoculture (52.00 ± 1.99) (Figure 1).

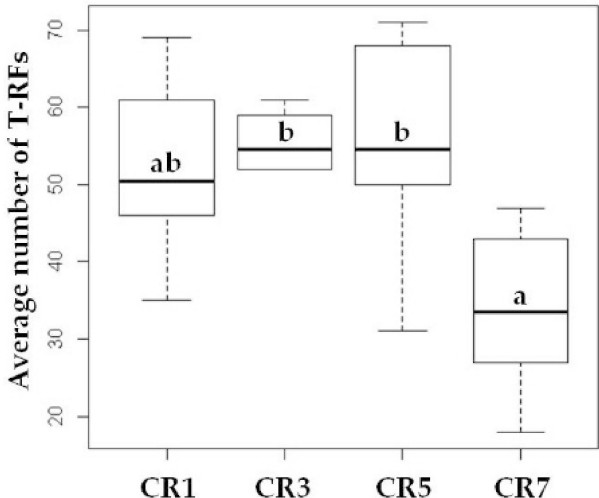

**Figure 1.** Box-plot representing the number of terminal restriction fragments (T-RFs) for each cropping system. Different letters indicate statistical differences between cropping systems according to one-way ANOVA combined with Tukey post-hoc test ($p < 0.05$).

Considering the estimated number of T-RFs (Chao1), the richness of the bacterial communities in CR5 ($p = 0.0224$) and CR1 ($p = 0.0324$) systems were significantly higher than that in CR7, but not that in CR3 ($p = 0.1302$). Biodiversity indices are summarized in Table 2. Based on the T-RFs, the overlap between bacterial communities is partly equivalent to the observation by Magurno et al. [22] working on arbuscular mycorrhizal fungi. The bacterial diversity represented by the Shannon Index H′ was significantly lower in the higher crop diversity systems (CR7) compared to the lower crop diversity systems (CR1: $p = 0.0478$; CR3: $p = 0.0263$; CR5: $p = 0.0208$). Similar to the studies of Soman et al. [38] and Peralta et al. [39], the differences in the structure of communities were not widely reflected in Shannon Index H′, indicating the necessity of using more indicators in order to understand differences between bacterial communities. The true richness and diversity, measured in estimated number of T-RFs, rather than Shannon Index H′, might better highlight the hypothetical influence of crop rotations on bacterial communities.

**Table 2.** Alpha diversity estimates of bacterial communities in soils of different cropping systems from the long-term experiment site Martonvásár.

| Cropping System | Estimated Number of T-RFs (Chao1) [1] | Evenness $e^H/S$ [1] | Shannon Index H′ [1] |
|---|---|---|---|
| CR1 | 91.88 ± 4.11 [b] | 0.70 ± 0.01 [a] | 3.57 ± 0.03 [b] |
| CR3 | 85.37 ± 1.98 [ab] | 0.70 ± 0.10 [a] | 3.65 ± 0.02 [b] |
| CR5 | 94.55 ± 5.72 [b] | 0.71 ± 0.01 [a] | 3.62 ± 0.03 [b] |
| CR7 | 47.67 ± 3.58 [a] | 0.79 ± 0.01 [a] | 3.22 ± 0.05 [a] |
| One-way ANOVA [2] | F = 4.462, $p = 0.0164$ * | F = 2.602, n.s. | F = 4.979, $p = 0.0109$ * |

[1] Values are presented as the mean ± SE, values within a column followed by different letters indicate statistical differences between treatments according to one-way ANOVA combined with Tukey post-hoc test (df = 3, $p < 0.05$);
[2] * Significant at $p < 0.05$, n.s. = nonsignificant.

Rotation has been widely considered as one of the most promising practices for the improvement of soil microbial diversity [39–41]. However, according to our results, the lowest diversity was found in the bacterial community associated to the most intense rotation system studied (Norfolk-type). This tendency correlates with the observations of Magurno et al. [22] in his study on arbuscular mycorrhizal fungi communities. The Norfolk-type rotation system could offer higher plant diversity but also causes more selective conditions compared to monoculture, as interpreted by Higo et al. [42]. The

daily average temperature was 26.9 °C in June, which is above the average of previous years, and the monthly average precipitation was 30% lower compared to previous years; this might affect our results. Although drought seems to have little impact on the bacterial diversity of soil communities, composition is significantly impacted by it [43].

T-RF numbers have been widely used as indicators of species richness in bacterial communities. However, an individual T-RF does not always represent an individual species or genus [44]. Therefore, it is important to note that relative phylotype richness and evenness indices extrapolated from T-RF data should be interpreted with a degree of caution. Navarro-Noya et al. [45] observed similar findings, identifying no significant differences in richness, diversity and total abundance between maize monoculture and wheat–maize rotation.

### 3.3. Differences in Bacterial Communities

Principal component analysis was performed with the data on T-RFs distribution as input to investigate more accurately the relationship among crop rotation systems and relative bacterial communities. PCA did not reveal a clear separation between maize monoculture and the other rotation systems (Figure 2). However, the pairwise tests of ANOSIM revealed that there are significant differences ($p = 0.0429$, $r = 0.3849$) in the composition of bacterial communities between the maize monoculture (CR1) and maize–alfalfa (CR3) rotation and between wheat–maize (CR5) and Norfolk-type rotation (CR7) ($p = 0.034$, $r = 0.3296$). The overall $p$ and R values were 0.0192 and 0.183, respectively.

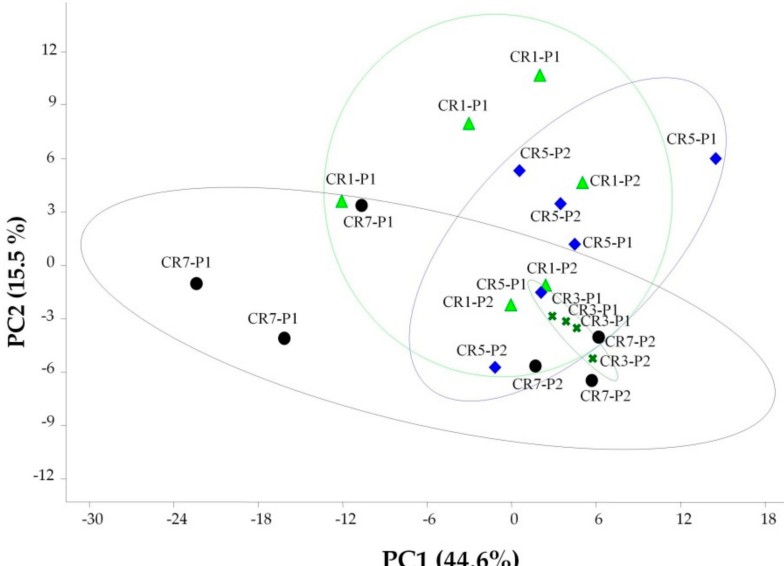

**Figure 2.** Two-dimensional plot of principal component analysis (PCA) based on the terminal restriction fragment length polymorphism (T-RFLP) dataset of the bacterial communities. Triangle: P1: subplot1; P2: subplot2. Numbers in parenthesis are variance percentage explained by each principal component (PC). Circles represent 70% confidence intervals for each type of sample separately calculated with the PAST program.

Changes in bacterial communities in response to crop rotations have been documented in several works [12,46–48], but opposing results have also been published [15,45]. While the phylotypes changed, the diversity of the community and the number of bacteria were similar in long-term crop rotation system compared to continuous monoculture in the work of Navarro-Noya et al. [45]. Silva et al. [49] confirmed that different crop rotations had only a minor influence on the composition of the bacterial community. The cultivated areas had different T-RFLP profiles compared to the noncultivated areas, and the diversity of the rRNA genes of α-proteobacteria, β-proteobacteria and *Actinobacteria* was reduced [50]. Maize and soybean rotation also change the composition of a microbial community; this

change is especially prominent in the number of Gram-positive bacteria [12]. Actively grown plants, in our case maize and wheat in Norfolk-type rotation, may interact differently with soil microbial communities; this can be seen in richness and community composition, as well as having been shown by Xin-ya et al. [51]. However, our findings are opposing since community richness was found to be lowest in sites planted with wheat [51]. Our results confirm the estimation of Hou et al. [52] that the rotation system could determine the microbial composition and biodiversity.

In some cases, the practice of rotation has a smaller impact on microbial diversity than other management practices, like NPK input as described by Zhao et al. [28]. Silva et al. [49] also confirmed that different crop rotations had only a minor influence on the composition of the bacterial community. Besides inducing changes in diversity, monoculture or short-term rotation has also altered, namely decreased, the microbial activity of soil, as shown in the studies of Hilton et al. [53] and Li et al. [54].

### *3.4. Relationship between Microbial Communities and Environmental Variables*

Canonical correlation analysis was performed to verify whether the differences in composition and structure of the bacterial communities could be due to the soil properties. According to the "Envfit" analysis, the strongest correlation was found with soil nitrogen, followed by soil pH and humus content. The effects of these soil properties on bacterial communities are shown by the direction and the length of the vectors (Figure 3).

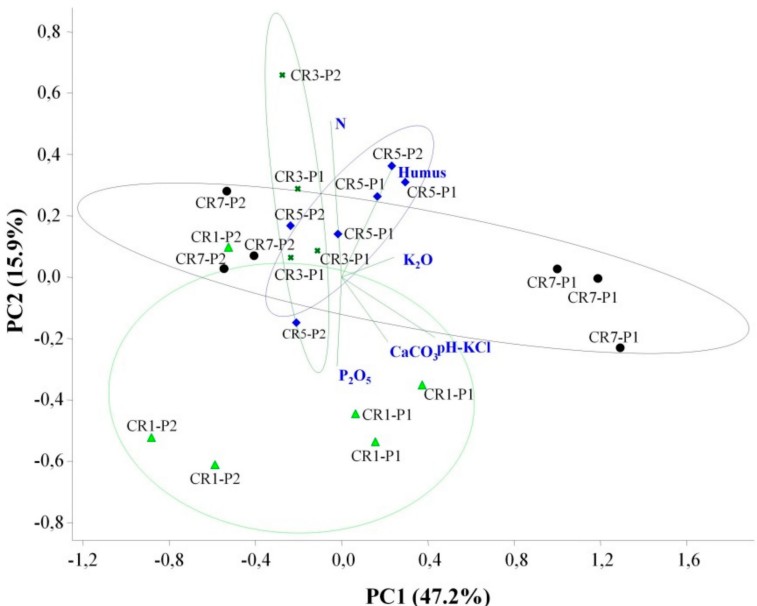

**Figure 3.** Ordination plot of principal component analysis (PCA) fitted with significant environmental vectors by "Envfit" script of R program.

Our results are consistent with several studies demonstrating that soil pH, nitrogen and humus belong to the strongest factors in structuring bacterial communities [27,55,56]. Crop rotations remarkably altered the microbial diversity, community composition and network, however it is highly dependent on the type of the given rotation system as well as the biotic and abiotic environmental factors [2,12].

### 4. Conclusions

The objective of this work was to give a snapshot of soil bacterial diversity and community structure in maize monoculture and three different crop rotations without identifying and classifying the community members. Long-term crop rotations have significant effects on soil humus content and P and N concentration, and moreover, they alter the composition of bacterial communities between

the maize monoculture and maize–alfalfa rotation as well as between wheat–maize and Norfolk-type rotations. Comparing the bio-markers and community changes, the microbial communities between different crop rotations should be explored in future studies for the improvement of sustainable agricultural productivity and plant protection. Therefore, in light of the present results it might be worth carrying out a deeper analysis.

**Author Contributions:** Conceptualization, Z.S. and K.P.; methodology, Z.M., Z.S. and K.P.; software, V.S. and B.V.; validation, Z.M. and B.V.; formal analysis, Z.M., Z.S. and K.P.; investigation, K.P.; resources, K.P.; data curation, Z.M., Z.S. and B.P.R.; writing—original draft preparation, Z.M., Z.S. and K.P.; writing—review and editing, K.P.; visualization, Z.M., V.S. and B.V.; supervision, K.P.; project administration, B.P.R.; funding acquisition, K.P.

**Funding:** This research was funded by grants from the National Research Council (OTKA K101878) and by NKFIH-1159-6/2019.

**Acknowledgments:** The authors thank Zoltán Berzsenyi and Péter Bónis for their valuable advice. We are grateful to Balázs Kovács for running the T-RFLP electrophoresis.

**Conflicts of Interest:** The authors declare no conflict of interest.

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
