# Peer review of "Effect of Long-Term Cropping Systems on the Diversity of the Soil Bacterial Communities"

_agronomy, doi:10.3390/agronomy9120878_

Round 1

Reviewer 1 Report

Line 38: correct to ‘activities’

Line 76-77: for each of the cropping seasons chosen, which plant species was actively growing at the time of sampling?  This may have an effect on bacterial diversity, as not all plants interact with soil microbiota with equal strength.

Line 79: is there a reason that June was selected for soil sampling? It is possible that sampling while plants are very young might not show much of an effect, if the plant is not interacting with soil microbiota much (similarly, a senesced plant wouldn’t interact with microbiota much and this might change the community composition of soil).

Lines 80-82: please describe how soil was collected: how much, how deep, were multiple cores taken and mixed for each sub plot?

Figure 2 and figure 3: what do P1 and P2 stand for in the plot labels?

In general, the paper is well written, but with so few samples, it is of limited scope.  There is definitely a lot of interest in the effect of cover crops on soil microbiota, however small the study.

  I was left with some questions about methods, as well as many speculative questions the authors may wish to include in the discussion: perhaps the rich soil type precludes the need for crops to recruit bacteria in that they have plenty of nutrients available already? Perhaps the crop species at the time of sampling was not interacting strongly with microbiota? If the soil was too hot and dry across all subplots, diversity might have been uniformly low (Ishaq et al. Geoderma 2020).

Author Response

We are greatly indebted to You for your corrections and comments on our manuscript entitled” Effect of long-term cropping systems on the diversity of the soil bacterial communities”

I have accepted all of your corrections, comments and let see my remarks and comments written in red. Please see the attachment "Responses to Reviewer 1Comments".

Thank you again for the most useful and helpful critical remarks!

Reviewer 2 Report

The mansucript is well prepared, however I would like to suggest to improve introduction section. The topic is new, however not very new, so It will be not so hard to find new references.

Author Response

We are greatly indebted to You for your corrections and comments on our manuscript entitled” Effect of long-term cropping systems on the diversity of the soil bacterial communities”

Following your suggesting, the Introduction part was improved using new references. Let see in revManuscript.

Thank you again for the most useful and helpful critical remarks!